# Evaluating Morphological Generalisation in Machine Translation by Distribution-Based Compositionality Assessment

**Anssi Moisio**
Department of Information and
Communications Engineering
Aalto University, Finland
anssi.moisio@aalto.fi

**Mathias Creutz**
Department of Digital Humanities
University of Helsinki, Finland
mathias.creutz@helsinki.fi

**Mikko Kurimo**
Department of Information and
Communications Engineering
Aalto University, Finland
mikko.kurimo@aalto.fi

## Abstract

Compositional generalisation refers to the ability to understand and generate a potentially infinite number of novel meanings using a finite group of known primitives and a set of rules to combine them. The degree to which artificial neural networks can learn this ability is an open question. Recently, some evaluation methods and benchmarks have been proposed to test compositional generalisation, but not many have focused on the morphological level of language. We propose an application of the previously developed *distribution-based compositionality assessment* method to assess morphological generalisation in NLP tasks, such as machine translation or paraphrase detection. We demonstrate the use of our method by comparing translation systems with different BPE vocabulary sizes. The evaluation method we propose suggests that small vocabularies help with morphological generalisation in NMT.[1]

## 1 Introduction

Natural languages usually adhere to the *principle of compositionality*, with the exception of idiomatic expressions. Partee et al. (1995) phrased this principle as "The meaning of a whole is a function of the meanings of the parts and of the way they are syntactically combined". Deriving from this principle, *compositional generalisation* (CG) refers to the capacity to understand and generate a potentially infinite number of novel meanings using a finite group of known primitives and a set of rules of how to combine them. In the case of language, morphemes are combined into words and words in turn into phrases and sentences, using the syntactical rules of the language.

Neural networks have long been argued to lack the ability to generalise compositionally the way humans do (Fodor and Pylyshyn, 1988; Marcus, 1998). After the rapid improvement of neural NLP systems during the previous decade, this question has gained renewed interest. Many new evaluation methods have been developed to assess whether the modern sequence-to-sequence (seq2seq) architectures such as Transformers exhibit CG, since they certainly exhibit increasingly competent linguistic behaviour. For instance, in one of the seminal CG evaluation methods, called SCAN (Lake and Baroni, 2018), a seq2seq system has seen certain natural language commands in training and needs to combine them in novel ways in testing.

CG is a general capacity that can be seen as a desideratum in many NLP tasks, and in machine learning more generally. Furthermore, CG is a multifaceted concept that can be, and should be, decomposed into narrower, more manageable aspects that can be tested separately (Hupkes et al., 2020). For example, NLP systems should be able to generalise compositionally both on the level of words and on the level of morphology.

Although many aspects of CG have recently been evaluated in NLP (an extensive review is offered by Hupkes et al. (2022)), some aspects have remained without an evaluation method. We identify (see Section 2) a lack of methods to

---

[1]Code and datasets available at https://github.com/anmoisio/morphogen-dbca

evaluate *compositional morphological* generalisation using only *natural*, non-synthetic, data. To fill this gap, we propose an application of the *distribution-based compositionality assessment* (DBCA) method (Keysers et al., 2020) (henceforth *Keysers*) to generate adversarial data splits to evaluate morphological generalisation in NLP systems.

Specifically, we split natural language corpora while controlling the distributions of lemmas and morphological features (*atoms* in the terminology of *Keysers*) on the one hand, and the distributions of the combinations of atoms (*compounds*, not to be confused with compound words) on the other hand. By requiring a low divergence between the atom distributions of the train and test sets, and a high divergence between the compound distributions, we can evaluate how well a system is able to generalise its morphological knowledge to unseen word forms.

For example, if our corpus included as atoms the lemmas "cat" and "dog", and the morphological tags `Number=Sing` and `Number=Plur`, a low divergence between the atom distributions would mean that both the training and test sets included all four of the atoms, and a high compound divergence would mean that the sets include different combinations of them, for instance training set {`cat, dogs`} and test set {`cats, dog`}.

Our main contributions are the following: **firstly**, we describe an application of DBCA to evaluate morphological generalisation in any NLP task in which the train and test data consist of sentences for which morphological tags are available. **Secondly**, we demonstrate how by this method we can evaluate morphological generalisation in machine translation without manual test design. And **thirdly**, using our proposed method, we assess the effect of the source language BPE (Sennrich et al., 2016) vocabulary size in Finnish-English NMT performance, and conclude that a smaller vocabulary helps the NMT models in morphological generalisation.

## 2 Background

In the broader field of machine learning, CG has been analysed in various domains besides that of natural language, such as visual question answering (Bahdanau et al., 2018), visual reasoning (Zerroug et al., 2022) and mathematics (Saxton et al., 2019), but in this work we focus on natural lan-

guage tasks. Two reviews have recently been published about CG in NLP, of which Donatelli and Koller (2023) focus on semantic parsing and the aforementioned Hupkes et al. (2022) (henceforth *Hupkes*) take a broader view, reviewing generalisation in general, not only the compositional type.

*Hupkes* categorised NLP generalisation experiments along five dimensions, of which we discuss two here to motivate our work. The first is the *type of generalisation* along which the *compositional* type is distinguished from the *morphological* type. *Hupkes* define compositionality as "the ability to systematically recombine previously learned elements to map new inputs made up from these elements to their correct output. In language, the inputs are 'forms' (e.g. phrases, sentences, larger pieces of discourse), and the output that they need to be mapped to is their meaning ...". In NMT, the translation works as a proxy to meaning, so that CG can be evaluated by evaluating the translation (Dankers et al., 2022) (other works that assess CG in NMT include (Li et al., 2021; Raunak et al., 2019)).

*Hupkes* contrast compositional with structural, including morphological, generalisation where an output space is not required but which focuses on generation of the correct forms. These definitions suggest a clear divide between the categories, which is understandable when analysing the literature: morphological generalisation, specifically inflection generation, has for decades been studied in psycholinguistics (Berko, 1958; Marcus et al., 1992) and computational linguistics (Rumelhart and McClelland, 1986; Corkery et al., 2019; Kodner et al., 2022). These studies do not address the question of how the different inflections are mapped to different meanings, hence they do not address *compositional* generalisation. However, inflections do bear meaning, of course, and so *compositional morphological* generalisation is an ability that humans possess, and NLP systems ought to be tested on.

Although *Hupkes* do not categorise any experiments as assessing *compositional morphological* generalisation, there has been at least one that we think could be so categorised: Burlot and Yvon (2017) designed an NMT test suite in which a single morphological feature is modified in a source language sentence, creating a contrastive pair, and the translations of the contrastive sentences are inspected for a corresponding change in the target

language.

The other dimension of *Hupkes* relevant to the motivation of our experiments is that of *shift source*: the shift between train and test sets could occur naturally (as in two natural corpora in different domains), it can be created by generating synthetic data, or an artificial partition of natural data can be obtained. Most of the previous methods to assess compositional generalisation in NMT (Burlot and Yvon, 2017; Li et al., 2021; Dankers et al., 2022) have synthetised data for the test sets. Generating synthetic data has its benefits: any morphological form can occur in the data when it is generated, and a single morphological feature can be easily focused on and evaluated qualitatively as well as quantitatively.

However, synthetic data has at least practical disadvantages, leaving aside the more theoretical question of how well the synthetic language approximates natural language, assuming the ultimate goal is systems that process natural language. In practice, synthetic test sets require manual design, which means it is difficult to come by a method to generate an unlimited number of synthetic sentences, or a method that could work in arbitrary languages. Furthermore, when manually designing test suites to evaluate morphological generalisation, as Burlot and Yvon (2017) designed, the requirement for manual work restricts the number of morphological phenomena we have resources to test.

The other option is to create artificial data splits of natural data. While natural data may be noisier and it might be more difficult to focus on a specific phenomenon of the language by this method, this method is easier to automate completely. Furthermore, the method of automatically generating data splits that we present in the next section is also generalisable to other tasks (e.g. paraphrase detection) and any corpus of sentences. Generating artificial data splits of natural data has previously been used to test CG in translation (Raunak et al., 2019), as well as to assess the capacity to capture long-distance dependencies in translation (Choshen and Abend, 2019), but not to assess *morphological* generalisation, as far as we are aware. (For a more general discussion of splitting data into non-random testing and training sets, see Søgaard et al. (2021).)

The method we describe in this paper is an application of the DBCA method developed by *Keysers*. Since this method is generic and task-agnostic, it can be applied to any dataset for which it is possible to define atom and compound distributions. Although it is easier to define these distributions for synthetic data, as in the CFQ dataset described by *Keysers*, it can also be applied to natural data, for example in semantic parsing (Shaw et al., 2021). The next section describes how DBCA can be used to assess morphological generalisation in any task where the training and testing corpora consist of natural language sentences.

## 3 Applying DBCA to assess morphological generalisation in NLP

DBCA is a method to evaluate CG by splitting a dataset into train/test sets with differing distributions, requiring some capacity to generalise from the training distribution to the test distribution. Specifically, the distributions of *atoms* (known primitives) and *compounds* (combinations of atoms) are controlled to get similar atom distributions but contrasting compound distributions in the training and test sets. In our application of DBCA to a corpus of natural language sentences, the atom distribution $\mathcal{F}_A$ of the corpus is the distribution of the lemmas and morphological features and the compound distribution $\mathcal{F}_C$ is the distribution of their combinations. Table 1 presents examples of atoms and compounds in this work.

To determine the atom and compound distributions, we first need to obtain the lemmas and morphological tags of all words in the corpus, which we accomplish for Finnish corpora using the Turku Neural Parser Pipeline (Kanerva et al., 2018). For the experiments presented in Section 4, we use a corpus of 1M sentences. In practice, we do not have resources to control the distribution of all lemmas even in this relatively small corpus, so we need to select some subset of the lemmas that we include in our analysis.

Selecting the lemma subset could be done in many ways, but the following is a way we deemed reasonable. To limit the number of lemmas, we first filter out lemmas that do not appear in the list of 94110 Finnish lemmas[2] or, since this list does not include proper names, in lists[3] of names

---

[2]Available at `https://kaino.kotus.fi/sanat/nykysuomi/`

[3]List of names of places: `https://kaino.kotus.fi/eksonyymit/?a=aineisto`
English given names: `https://en.wiktionary.org/wiki/Appendix:English_given_names`

| | Atoms | Compounds |
|---|---|---|
| Desc. | lemmas and morphological tags | combinations of atoms |
| E.g. | `tunturi,` `Case=Gen,` `Case=Ade,` `Number=Sing,` `Number=Plur` | `tunturi|Case=Gen|Number=Plur` (*tunturien*, of mountains); `tunturi|Case=Ade|Number=Sing` (*tunturilla*, on mountain) |

Table 1: Description and examples of what we call "atoms" and "compounds". The compounds are the unique word forms, determined by the lemma and the morphological tags. The word form and its English translation are written inside the brackets.

for places, or lists of Finnish and English given names. This way, the lemmas that are filtered out include most of the typos and other non-words. Then we rank the remaining lemmas by frequency in our corpus, and sample a fixed number of lemma occurrences from constant intervals in the ranked list of lemmas. Specifically, we take 40000 lemma occurrences at intervals of 1000 lemma types in the list of lemmas. For our corpus of 1M sentences, this method subsamples the lemmas with frequency ranks of 1000-1033, 2000-2083, 3000-3174, and so on, so that there are fewer frequent lemma types than rare lemma types, but the total number of occurrences of each bucket is around 40k. Lemmas that occur fewer than 10 times in the corpus are excluded. After the filtering, we have 8720 lemma types that occur about 390k times in total in our corpus of 1M sentences. We append the list of 48 morphological tags[4] (after filtering some that indicate uninteresting words such as 'Typo' and 'Abbr') that these lemmas appear with to the lemma list to complete our list of atoms.

*Keysers* weighted the compounds to "avoid double-counting compounds that are highly correlated with some of their super-compounds". The idea is to lessen the weight of those compounds that only or often occur as a part of one certain super-compound. We weight the compounds analogously, but use only two levels in our weighting, which makes the weighting simpler than in *Keysers*: we consider the combinations of morphological tags as the lower level of compounds, and these combined with lemmas as the higher level. Thus the motivation for weighting in our case is not to use those morphological tag combinations that only occur with some specific lemma. Therefore, we look for the lemma with

which each morphological tag combination occurs most often, and give the tag combination a weight that is the complement of the empirical probability that the tag combination occurs with this lemma. For example, we found that the rare morph tag combination `Case=Ade | Degree=Pos | Number=Plur | PartForm=Pres | VerbForm=Part | Voice=Pass` occurs 84% of the time with the lemma `saada` forming the word "saatavilla", so it gets a weight of 0.16. After weighting the tag combinations, we exclude those that have a weight of 0.33 or less.

After the described filtering steps, we have 8322 atoms, which includes the lemmas and morphological tags. The atoms occur about 1.3M times in 273k sentences in our corpus of 1M sentences. There are 335 morphological tag combinations, which create about 69k unique word forms with the lemmas; i.e. we use 69k compounds in our analysis. These compounds occur 352k times in the corpus.

Calculating atom and compound divergences is done the same way as in *Keysers*. Namely, divergence $\mathcal{D}$ between distributions $P$ and $Q$ is calculated using the Chernoff coefficient $C_\alpha(P\|Q) = \sum_k p_k^\alpha q_k^{1-\alpha} \in [0,1]$ (Chung et al., 1989), with $\alpha = 0.5$ for the atom divergence and $\alpha = 0.1$ for the compound divergence. As described by *Keysers*, $\alpha = 0.5$ for the atom divergence "reflects the desire of making the atom distributions in train and test as similar as possible", and $\alpha = 0.1$ for the compound divergence "reflects the intuition that it is more important whether a certain compound occurs in P (train) than whether the probabilities in P (train) and Q (test) match exactly". Since the Chernoff coefficient is a similarity metric, the atom and compound divergences of a train set $V$ and a test set $W$ are:

$$\mathcal{D}_A(V\|W) = 1 - C_{0.5}(\mathcal{F}_A(V) \| \mathcal{F}_A(W))$$
$$\mathcal{D}_C(V\|W) = 1 - C_{0.1}(\mathcal{F}_C(V) \| \mathcal{F}_C(W)).$$

---

and Finnish: `https://tinyurl.com/3mn52ms6` `https://tinyurl.com/mwjvaxkk`

[4]See `https://universaldependencies.org/docs/fi/feat/` for the list of Finnish morphological tags.

**Procedure 1** Data division algorithm.

**Input:** $G$           ▷ Corpus of sentences
**Input:** $N$        ▷ Use $N$ sentences from $G$
**Input:** $a$      ▷ Lower bound for $|V|/|W|$
**Input:** $b$      ▷ Upper bound for $|V|/|W|$
**Output:** $V, W$     ▷ Train set, test set
   $V \leftarrow \{x \in_R G\}$      ▷ A random sentence
   $W \leftarrow \emptyset$
   $G \leftarrow G \backslash V$
   **for** $i \leftarrow 1$ **to** $N$ **do**
      $r \leftarrow |V|/|W|$
      $s_V \leftarrow \max_{x \in G} \text{score}(V \cup \{x\}, W)$
      $i_V \leftarrow \text{argmax}_{x \in G} \text{score}(V \cup \{x\}, W)$
      $s_W \leftarrow \max_{x \in G} \text{score}(V, W \cup \{x\})$
      $i_W \leftarrow \text{argmax}_{x \in G} \text{score}(V, W \cup \{x\})$
      **if** $(s_V > s_W \wedge r < b) \vee r < a$ **then**
         $V \leftarrow V \cup \{i_V\}$
         $G \leftarrow G \backslash \{i_V\}$
      **else**
         $W \leftarrow W \cup \{i_W\}$
         $G \leftarrow G \backslash \{i_W\}$
      **end if**
   **end for**

Once the divergences are defined, we can split a corpus of natural language sentences into training and testing sets with an arbitrary compound and atom divergence values. For this, we use a simple greedy algorithm, sketched in Algorithm 1. For a maximum compound divergence split, the score is calculated as

$$\text{score}(Q, P) = \mathcal{D}_C(Q\|P) - \mathcal{D}_A(Q\|P),$$

and in general, for any desired compound divergence value $c$:

$$\text{score}(Q, P) = -|c - \mathcal{D}_C(Q\|P)| - \mathcal{D}_A(Q\|P).$$

In practice, we do not have resources to calculate the $\max_{x \in G}$ score. Instead, at each iteration we take a subset $G' \subset G$, say 1000 sentences, and calculate $\max_{x \in G'}$ score.

As mentioned above, this method can be used for any corpus that consists of natural language sentences for which the morphological tags can be obtained. In the next section we use this method to assess morphological generalisation in machine translation.

## 4 Experiments and results

### 4.1 NMT model training setup and data

We chose Finnish as the language we analyse because of its rich morphology and because there is a good morphological tagger available for Finnish. We use the English-Finnish parallel corpus from the Tatoeba challenge data release (Tiedemann, 2020). We first apply some heuristics provided by Aulamo et al. (2020) to remove noisy data, and restrict the maximum sentence length to 100 words, after which we take a random sample of 1 million sentence pairs.

We use the OpenNMT-py (Klein et al., 2017) library to train Finnish-English Transformer NMT models using the hyperparameters provided in the example config file[5], which includes the standard 6 transformer layers with 8 heads and a hidden dimension of 512, as in (Vaswani et al., 2017). We train the models until convergence or until a maximum of 33000 steps with 2000 warm-up steps and a batch size of 4096 tokens.

For more details about the setup, see the Github repository linked on the first page.

### 4.2 The effect of compound divergence on translation performance

The basic experiment we propose is to make at least two different train/test splits of a corpus, using $\mathcal{D}_C$ values of 0 and 1, respectively, (keeping $\mathcal{D}_A = 0$) and assess the change in translation performance (for which we use BLEU (Papineni et al., 2002) and chrF2++ (Popović, 2017) as metrics). Since with $\mathcal{D}_C = 1$ there are more unseen word forms in the test set, we expect a decrease in translation performance from $\mathcal{D}_C = 0$ to $\mathcal{D}_C = 1$ that is caused by the $\mathcal{D}_C = 1$ test set requiring more morphological generalisation capacity.

We show empirically the decrease in performance in Section 4.3, but the cause of this decrease is of course more difficult to verify exactly. The atom and compound distributions are the only things we explicitly control when splitting the corpus, and we only require the compound divergence to differ between different data splits. Therefore, we assume the differing compound divergence to be the cause of this effect, but to be more certain, we conduct two simple checks to look for confounding factors.

---

[5]https://github.com/OpenNMT/OpenNMT-py/blob/9d617b8b/config/config-transformer-base-1GPU.yml

Firstly, an increase in the average sentence length could be another factor that makes one test set more difficult than another. Increasing the sequence length from training to test set is actually a method that has been proposed to test a certain type of compositional generalisation, sometimes called *productivity* (Hupkes et al., 2020; Raunak et al., 2019). We calculated the average sentence lengths of the train and test sets of the 8 different data splits that we obtained using 8 different random seeds for the data split algorithm. What we found is that for $\mathcal{D}_C = 1$ the average lengths in test sets are actually shorter (ranging from 11.35 to 11.66 words) than those for $\mathcal{D}_C = 0$ (ranging from 12.27 to 13.72 words). The average training set sentence lengths are similar for both $\mathcal{D}_C$ values, ranging from 8.66 to 8.79 for $\mathcal{D}_C = 0$ and from 8.65 to 8.73 for $\mathcal{D}_C = 1$. Thus we know that an increased difference between train and test set sentence lengths cannot explain the decrease in NMT performance from $\mathcal{D}_C = 0$ to $\mathcal{D}_C = 1$ since the difference is actually larger for $\mathcal{D}_C = 0$. The fact that the average sentence length in training sets is always significantly shorter than in test sets is an interesting unintended artefact of the data division algorithm that deserves further investigation in the future, but it does not confound our analysis.

As the second sanity check, we evaluated the NMT models on a neutral test set to see if, for any reason, the training set would be in general worse with $\mathcal{D}_C = 1$ than with $\mathcal{D}_C = 0$, instead of only being worse for the specific test set that we have created. For this we used the Tatoeba challenge test set, which we did not use to train or tune the hyperparameters of any models. The results for the vocabulary size 1000 are presented in Figure 1. We used the models trained on the training sets from the data splits with compound divergences 0.0, 0.5 and 1.0. The compound divergences between these training sets and the Tatoeba challenge test set do correlate with the target $\mathcal{D}_C$ of the data split, but they range only from about 0.4 to 0.6.

From Figure 1 we can see that the NMT models trained with different data sets, from data splits with different $\mathcal{D}_C$ values, do not show similar decrease in performance on the neutral-ish Tatoeba challenge test set as on the test sets obtained from the data split algorithm. We take this to mean that the models trained on $\mathcal{D}_C = 1$ data splits are not in general worse than those trained with $\mathcal{D}_C = 0$

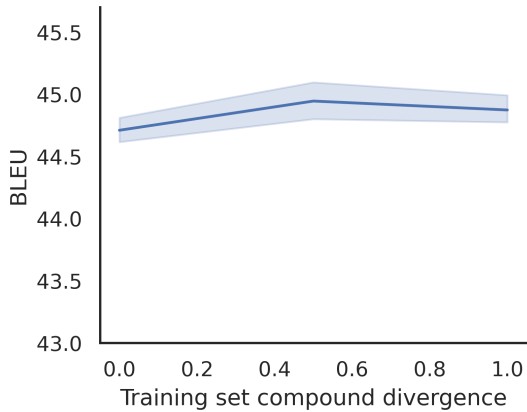

Figure 1: Results on the Tatoeba challenge test set. The x-axis labels denote the compound divergences between the training sets and the test sets analysed later in Figure 2. That is, the divergence is not between the training sets and the Tatoeba challenge test set.

data splits, but only worse on the high-divergence test set.

### 4.3 The effect of BPE vocabulary size on morphological generalisation in NMT

Next, we make the assumption, based on the analysis in Section 4.2, that we can measure morphological generalisation by measuring the decrease of NMT performance between train/test splits of $\mathcal{D}_C = 0$ and $\mathcal{D}_C = 1$. Previous studies have suggested the hypothesis that NMT models with smaller BPE vocabularies are more capable of modelling morphological phenomena than those with larger vocabularies (for example Libovický and Fraser (2020)). In this section, we compare the morphological generalisation capacities of NMT models with different source-side (Finnish) vocabulary sizes, using the method we have proposed.

As a preliminary experiment, we tuned the BPE vocabulary size for our setup (see Section 4.1) on the Tatoeba challenge development set, and found the optimal size to be around 3000 BPE tokens for both the source and target languages. Since we are interested in the Finnish morphology, next we kept the target (English) vocabulary size constant and varied only the source-side vocabulary size.

One thing to note about the vocabulary size is that when we train an NMT system keeping the number of tokens in each batch constant, the number of steps until convergence usually decreases

when the vocabulary size increases, since one epoch takes fewer steps. This reduction in compute, when using a larger vocabulary, is to some extent compensated by the increase of the input layer size (and output layer size, if target language vocabulary is increased too).

We chose 7 different vocabulary sizes, 3 larger and 3 smaller than the optimal 3000, and evaluated them with target compound divergence values of 0.0, 0.25, 0.5, 0.75 and 1.0. The sizes of the test sets are in the order of a few tens of thousands, or a little over a hundred thousand, sentences. The relatively large test set size leads to statistical significance even for small BLEU differences (see Table 3 for details).

From the BLEU results for $\mathcal{D}_C = 0$ and $\mathcal{D}_C = 1$ in Figure 2 we can see that the BLEU results drop, as expected, when the test set demands (more) capacity to generalise to unseen morphological forms. Furthermore, when comparing the different vocabulary sizes, we can notice that as we either increase or decrease the vocab size from 3000, the performance drops, but it drops slightly differently w.r.t $\mathcal{D}_C$. This effect is most conspicuous for the pair of sizes 500 and 18000. The larger vocabulary performs slightly better when there is less need for morphological generalisation, but the small vocabulary performs better when it is needed more. In general, from this figure we can see that the vocabulary size roughly correlates with the angle of the downward slope, suggesting that the larger the vocabulary, the poorer the capacity for morphological generalisation.

To investigate the effect of the initialisation of the data split algorithm on the results, we split the same corpus starting from 8 different random initialisations, and trained NMT models for each data split. For this, we chose two pairs of vocabulary sizes that showed most clearly contrasting performance w.r.t $\mathcal{D}_C$: 500&18000 and 1000&6000. The main results are presented in Table 2. For these results, the test sets of the 8 random seeds are concatenated together to create exceptionally large test sets of around 400k-500k sentences. The results for the individual data splits are presented in Appendix A in Table 4.

From these results we can see the same contrasting performance of the small and large vocabularies w.r.t the different compound divergence values. The difference is small but statistically significant. The models with small vocabularies show better

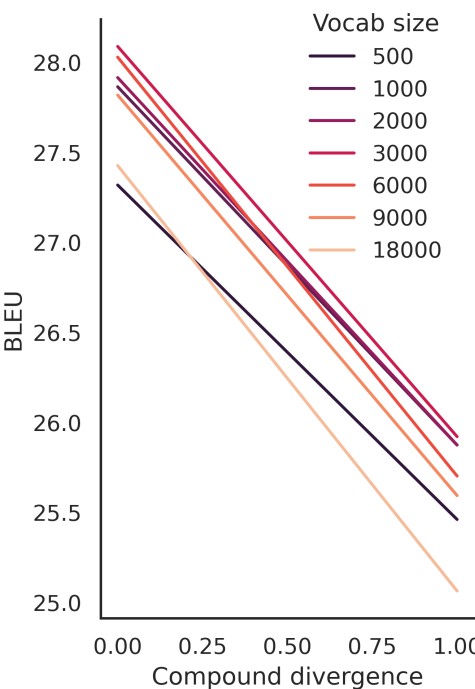

Figure 2: Different source vocabulary sizes evaluated with minimum and maximum (0 and 1) compound divergence data splits. Compound divergence value 1 requires more morphological generalisation. The larger the vocabulary the steeper the slope, suggesting poorer ability to generalise. For more details, see Table 3 in Appendix A.

performance than those with large ones when morphological generalisation is needed, and vice versa when morphological generalisation is not needed as much.

## 5 Discussion and future work

In Section 3, we proposed an application of DBCA to divide any corpus of sentences, for which morphological tags are available, into training and test sets with similar distributions of lemmas and morphological tags but contrasting distributions of word forms, in order to assess morphological generalisation. By this method, we can take a large proportion of the morphological phenomena of a selected language into consideration, in our experiments 335 different morphological categories that together with about 8k lemmas create 69k unique Finnish word forms, and evaluate the effects of the contrasting train/test distributions of the word forms in machine translation. This enables a different, complementing type of assessment of morphological generalisation than previous synthetic

| | chrF2++ | | BLEU | |
| Vocab | $\mathcal{D}_C = 0$ | $\mathcal{D}_C = 1$ | $\mathcal{D}_C = 0$ | $\mathcal{D}_C = 1$ |
|---|---|---|---|---|
| 500 | 51.20 (51.20 ± 0.05) | **49.33** (49.33 ± 0.05) | 27.50 (27.50 ± 0.07) | **25.4** (25.40 ± 0.07) |
| 18000 | **51.29** (51.29 ± 0.05) | 49.04 (49.05 ± 0.05) | **27.69** (27.69 ± 0.07) | 25.18 (25.18 ± 0.07) |
| | $p = 0.0003$ | $p = 0.0003$ | $p = 0.0003$ | $p = 0.0003$ |
| 1000 | 51.78 (51.78 ± 0.05) | **49.79** (49.79 ± 0.05) | 28.17 (28.17 ± 0.07) | **25.89** (25.89 ± 0.07) |
| 6000 | **51.83** (51.83 ± 0.05) | 49.67 (49.67 ± 0.05) | **28.24** (28.24 ± 0.07) | 25.80 (25.80 ± 0.07) |
| | $p = 0.0003$ | $p = 0.0003$ | $p = 0.0003$ | $p = 0.0003$ |

Table 2: Pairwise comparisons of the source vocabulary sizes 500 and 18000; 1000 and 6000. The results are calculated for the concatenated test sets generated with 8 random seeds. Inside brackets is the true mean estimated from bootstrap resampling and the 95% confidence interval. The results for the individual seeds are presented in Appendix A in Table 4 and Figure 3.

benchmarks (mainly Burlot and Yvon (2017)) that focus on a smaller number of morphological phenomena. One benefit of our method is its comprehensiveness, focusing on the corpus-wide distributions of word forms.

Using only corpus-wide metrics such as BLEU, as we used, does not discriminate between the morphological errors, which we are interested in, and other kinds of translation errors. In the terminology of Burlot and Yvon (2017), this *holistic*, document-level evaluation can be contrasted with *analytic* evaluation that focuses more specifically on difficulties in morphology. A trick that could enable a more analytic assessment of the translations of the unseen word forms would be to align the words in the source sentences with the words in the reference translations and the words in the predicted translations, and evaluate only the translations of the parts of the sentences that correspond to the unseen word forms. Similar method has been used previously for example by Bau et al. (2019); Stanovsky et al. (2019).

Especially combined with this word-alignment trick, we could also make our evaluation more *fine-grained* (this concept also from Burlot and Yvon (2017)), that is, our evaluation could differentiate between different types of mistakes. Since we have the morphological tags, we could sort the words by morphological category and compare the translation accuracies to look for any especially difficult categories for the translation models.

To demonstrate the use of our proposed method, we compared NMT models with different BPE vocabulary sizes, since vocabulary size has been hypothesised to affect the capacity to model morphology in translation. Besides vocabulary size,

there are many other model design choices that have been proposed to help either in generalisation or in capturing morphological phenomena. Tokenisation methods that are more linguistically motivated than BPE, such as the Morfessor methods (Creutz and Lagus, 2002; Virpioja et al., 2013) or LMVR (Ataman et al., 2017), should help with morphological generalisation since the tokens produced by these methods approximate the linguistic morphemes more closely. Factored NMT systems (García-Martínez et al., 2016) can cover more of the target side vocabulary than subword-based NMT systems, which can also help in modelling the morphology of the target language. We hope our evaluation method will help assessing alternative NMT methods, such as these, from the perspective of morphological generalisation.

The DBCA method is general, and could be applied to a wide variety of tasks and datasets. Our application of DBCA is more specific, but it still inherits some of the generality of the original method. Our method is directly applicable to any machine learning task in which the dataset consists of sentences for which the morphological tags are available. In the future, we intend to extend our assessment of morphological generalisation to other languages, as well as to other NLP tasks, such as paraphrase detection.

## 6 Conclusion

We proposed a method to assess morphological generalisation by distribution-based compositionality assessment. Because this method is fully automated, it enables more comprehensive assessment of morphological generalisation than previously proposed synthetic benchmarks, in terms

of the number of inflection types we can evaluate. We used our method to assess NMT models with different BPE vocabulary sizes and found that models with smaller vocabularies are better at morphological generalisation than those with larger vocabularies. Lastly, we discussed the varied future directions that our generalisable method offers, such as assessing morphological generalisation in other NLP tasks besides NMT.

# 7 Acknowledgements

We thank Jörg Tiedemann, Eetu Sjöblom and others in the Helsinki Language Technology and Aalto Speech Recognition research groups for helpful discussions and advice. We also thank the anonymous reviewers for their insightful comments and feedback. The work was supported by the Academy of Finland grant 337073. The computational resources were provided by Aalto ScienceIT.

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

# A   Detailed results

Table 3 lists the results for the different source-side (Finnish) BPE vocabulary sizes and different compound divergence values. Table 4 includes the pairwise comparisons of vocabulary sizes 500 and 18000 and 1000 and 6000 for all random seeds. Figure 3 presents in a graph the pairwise comparisons of vocabulary sizes 500 and 18000, with all compound divergence values.

| Vocab size | BLEU per compound divergence | | | | |
|---|---|---|---|---|---|
| | 0.0 | 0.25 | 0.5 | 0.75 | 1.0 |
| 500 | 27.32 (27.32 ± 0.17) | 26.57 (26.57 ± 0.20) | 25.36 (25.35 ± 0.17) | 24.77 (24.76 ± 0.18) | 25.46 (25.46 ± 0.17) |
| 1000 | 27.86 (27.87 ± 0.18) | 27.33 (27.33 ± 0.20) | 25.87 (25.87 ± 0.18) | 25.56 (25.55 ± 0.18) | 25.87 (25.87 ± 0.18) |
| 2000 | 27.91 (27.92 ± 0.18) | 27.58 (27.58 ± 0.20) | 26.07 (26.07 ± 0.18) | 25.53 (25.53 ± 0.18) | 25.87 (25.88 ± 0.17) |
| 3000 | 28.09 (28.09 ± 0.18) | 27.54 (27.54 ± 0.20) | 25.98 (25.97 ± 0.17) | 25.69 (25.69 ± 0.18) | 25.92 (25.92 ± 0.18) |
| 6000 | 28.03 (28.03 ± 0.18) | 27.37 (27.36 ± 0.20) | 25.98 (25.98 ± 0.18) | 25.44 (25.44 ± 0.19) | 25.70 (25.70 ± 0.17) |
| 9000 | 27.82 (27.82 ± 0.19) | 27.26 (27.26 ± 0.21) | 25.73 (25.73 ± 0.17) | 25.36 (25.36 ± 0.19) | 25.59 (25.59 ± 0.18) |
| 18000 | 27.43 (27.43 ± 0.18) | 26.81 (26.81 ± 0.21) | 25.36 (25.35 ± 0.17) | 24.74 (24.74 ± 0.19) | 25.06 (25.06 ± 0.17) |
| | chrF2++ per compound divergence | | | | |
| 500 | 51.01 (51.01 ± 0.14) | 50.58 (50.58 ± 0.16) | 49.75 (49.75 ± 0.14) | 49.24 (49.24 ± 0.16) | 49.19 (49.19 ± 0.14) |
| 1000 | 51.53 (51.53 ± 0.14) | 51.33 (51.33 ± 0.16) | 50.30 (50.30 ± 0.14) | 49.98 (49.98 ± 0.15) | 49.59 (49.59 ± 0.14) |
| 2000 | 51.54 (51.54 ± 0.14) | 51.52 (51.52 ± 0.16) | 50.40 (50.40 ± 0.14) | 49.91 (49.91 ± 0.15) | 49.68 (49.68 ± 0.14) |
| 3000 | 51.68 (51.69 ± 0.14) | 51.47 (51.47 ± 0.16) | 50.40 (50.40 ± 0.14) | 50.04 (50.04 ± 0.15) | 49.62 (49.62 ± 0.14) |
| 6000 | 51.66 (51.66 ± 0.14) | 51.33 (51.33 ± 0.16) | 50.32 (50.32 ± 0.14) | 49.79 (49.79 ± 0.16) | 49.48 (49.48 ± 0.14) |
| 9000 | 51.37 (51.37 ± 0.14) | 51.09 (51.09 ± 0.16) | 50.07 (50.07 ± 0.14) | 49.78 (49.77 ± 0.16) | 49.36 (49.36 ± 0.14) |
| 18000 | 51.02 (51.03 ± 0.14) | 50.78 (50.78 ± 0.16) | 49.74 (49.74 ± 0.14) | 49.23 (49.23 ± 0.15) | 48.78 (48.78 ± 0.14) |

Table 3: The BLEU and chrF2++ results for the different source-side (Finnish) BPE vocabulary sizes and different compound divergence values. Inside brackets is the true mean estimated from bootstrap resampling and the 95% confidence interval.

| | | chrF2++ | | BLEU | |
|---|---|---|---|---|---|
| Seed | Vocab | $\mathcal{D}_C = 0$ | $\mathcal{D}_C = 1$ | $\mathcal{D}_C = 0$ | $\mathcal{D}_C = 1$ |
| 11 | 500 | 51.01 (51.01 ± 0.14) | 49.19 (49.19 ± 0.14) | 27.32 (27.32 ± 0.17) | 25.46 (25.46 ± 0.17) |
| | 18000 | 51.02 (51.03 ± 0.14) | 48.78 (48.78 ± 0.14) | 27.43 (27.43 ± 0.18) | 25.06 (25.06 ± 0.17) |
| | | p = 0.2439 | p = 0.0003 | p = 0.0243 | p = 0.0003 |
| 22 | 500 | 51.01 (51.01 ± 0.14) | 49.08 (49.08 ± 0.15) | 27.3 (27.3 ± 0.18) | 25.2 (25.2 ± 0.18) |
| | 18000 | 50.85 (50.85 ± 0.14) | 49.05 (49.05 ± 0.15) | 27.17 (27.17 ± 0.18) | 25.1 (25.1 ± 0.18) |
| | | p = 0.0003 | p = 0.1913 | p = 0.0107 | p = 0.053 |
| 33 | 500 | 51.07 (51.07 ± 0.14) | 49.37 (49.37 ± 0.17) | 27.37 (27.37 ± 0.18) | 25.09 (25.09 ± 0.2) |
| | 18000 | 50.97 (50.97 ± 0.14) | 49.04 (49.04 ± 0.17) | 27.3 (27.3 ± 0.18) | 24.83 (24.83 ± 0.2) |
| | | p = 0.0047 | p = 0.0003 | p = 0.092 | p = 0.0003 |
| 44 | 500 | 52.02 (52.02 ± 0.17) | 49.7 (49.7 ± 0.18) | 28.3 (28.3 ± 0.21) | 25.8 (25.8 ± 0.22) |
| | 18000 | 52.44 (52.44 ± 0.17) | 49.43 (49.43 ± 0.17) | 28.72 (28.72 ± 0.21) | 25.63 (25.63 ± 0.22) |
| | | p = 0.0003 | p = 0.0003 | p = 0.0003 | p = 0.0077 |
| 55 | 500 | 52.33 (52.34 ± 0.18) | 49.34 (49.34 ± 0.16) | 29.04 (29.04 ± 0.23) | 25.29 (25.29 ± 0.2) |
| | 18000 | 52.76 (52.76 ± 0.18) | 49.04 (49.04 ± 0.16) | 29.58 (29.58 ± 0.24) | 25.08 (25.08 ± 0.2) |
| | | p = 0.0003 | p = 0.0003 | p = 0.0003 | p = 0.001 |
| 66 | 500 | 50.98 (50.98 ± 0.14) | 49.24 (49.24 ± 0.14) | 27.12 (27.12 ± 0.18) | 25.31 (25.31 ± 0.18) |
| | 18000 | 51.06 (51.06 ± 0.14) | 48.87 (48.87 ± 0.14) | 27.4 (27.4 ± 0.18) | 25.04 (25.04 ± 0.17) |
| | | p = 0.0183 | p = 0.0003 | p = 0.0003 | p = 0.0003 |
| 77 | 500 | 50.84 (50.83 ± 0.14) | 49.46 (49.46 ± 0.14) | 27.12 (27.12 ± 0.18) | 25.41 (25.4 ± 0.16) |
| | 18000 | 50.68 (50.68 ± 0.14) | 49.22 (49.22 ± 0.14) | 27.06 (27.06 ± 0.18) | 25.25 (25.25 ± 0.17) |
| | | p = 0.0007 | p = 0.0003 | p = 0.1186 | p = 0.0023 |
| 88 | 500 | 50.97 (50.97 ± 0.14) | 49.38 (49.38 ± 0.14) | 27.22 (27.22 ± 0.18) | 25.61 (25.61 ± 0.17) |
| | 18000 | 51.37 (51.37 ± 0.14) | 49.05 (49.05 ± 0.14) | 27.81 (27.81 ± 0.18) | 25.43 (25.43 ± 0.18) |
| | | p = 0.0003 | p = 0.0003 | p = 0.0003 | p = 0.0003 |
| 11 | 1000 | 51.53 (51.53 ± 0.14) | 49.59 (49.59 ± 0.14) | 27.86 (27.87 ± 0.18) | 25.87 (25.87 ± 0.18) |
| | 6000 | 51.66 (51.66 ± 0.14) | 49.48 (49.48 ± 0.14) | 28.03 (28.03 ± 0.18) | 25.7 (25.7 ± 0.17) |
| | | p = 0.0003 | p = 0.0017 | p = 0.0013 | p = 0.001 |
| 22 | 1000 | 51.46 (51.46 ± 0.14) | 49.64 (49.64 ± 0.15) | 27.9 (27.9 ± 0.18) | 25.69 (25.69 ± 0.18) |
| | 6000 | 51.47 (51.47 ± 0.14) | 49.61 (49.61 ± 0.15) | 27.94 (27.94 ± 0.19) | 25.64 (25.64 ± 0.18) |
| | | p = 0.3059 | p = 0.1786 | p = 0.1519 | p = 0.1383 |
| 33 | 1000 | 51.59 (51.59 ± 0.14) | 49.7 (49.7 ± 0.17) | 27.89 (27.88 ± 0.18) | 25.45 (25.45 ± 0.2) |
| | 6000 | 51.63 (51.63 ± 0.14) | 49.67 (49.68 ± 0.17) | 28.02 (28.02 ± 0.18) | 25.51 (25.51 ± 0.21) |
| | | p = 0.117 | p = 0.2073 | p = 0.0047 | p = 0.1276 |
| 44 | 1000 | 52.67 (52.67 ± 0.16) | 50.32 (50.32 ± 0.17) | 29.01 (29.01 ± 0.21) | 26.53 (26.53 ± 0.22) |
| | 6000 | 52.68 (52.68 ± 0.16) | 50.06 (50.06 ± 0.18) | 29.01 (29.01 ± 0.22) | 26.33 (26.33 ± 0.22) |
| | | p = 0.2809 | p = 0.0003 | p = 0.3949 | p = 0.0037 |
| 55 | 1000 | 52.8 (52.8 ± 0.18) | 49.92 (49.92 ± 0.16) | 29.66 (29.66 ± 0.24) | 25.92 (25.92 ± 0.2) |
| | 6000 | 53.02 (53.03 ± 0.18) | 49.72 (49.73 ± 0.16) | 29.84 (29.85 ± 0.24) | 25.73 (25.73 ± 0.2) |
| | | p = 0.0003 | p = 0.0003 | p = 0.0017 | p = 0.0003 |
| 66 | 1000 | 51.39 (51.39 ± 0.14) | 49.57 (49.57 ± 0.14) | 27.64 (27.64 ± 0.18) | 25.71 (25.71 ± 0.18) |
| | 6000 | 51.5 (51.49 ± 0.14) | 49.37 (49.37 ± 0.14) | 27.79 (27.79 ± 0.19) | 25.54 (25.54 ± 0.18) |
| | | p = 0.0013 | p = 0.0003 | p = 0.0017 | p = 0.0017 |
| 77 | 1000 | 51.51 (51.51 ± 0.15) | 49.8 (49.8 ± 0.13) | 27.86 (27.86 ± 0.18) | 25.84 (25.84 ± 0.17) |
| | 6000 | 51.76 (51.76 ± 0.14) | 49.74 (49.74 ± 0.14) | 28.09 (28.09 ± 0.19) | 25.74 (25.74 ± 0.17) |
| | | p = 0.0003 | p = 0.0453 | p = 0.0003 | p = 0.022 |
| 88 | 1000 | 51.9 (51.9 ± 0.14) | 49.95 (49.95 ± 0.14) | 28.29 (28.29 ± 0.18) | 26.2 (26.2 ± 0.18) |
| | 6000 | 51.6 (51.6 ± 0.14) | 49.84 (49.84 ± 0.14) | 28.01 (28.01 ± 0.18) | 26.23 (26.23 ± 0.18) |
| | | p = 0.0003 | p = 0.0007 | p = 0.0003 | p = 0.2209 |

Table 4: Pairwise comparisons of the source vocabulary sizes 500 and 18000; 1000 and 6000 on the minimum and maximum compound divergence data splits. For 8 data split algorithm random seeds. Inside brackets is the true mean estimated from bootstrap resampling and the 95% confidence interval.

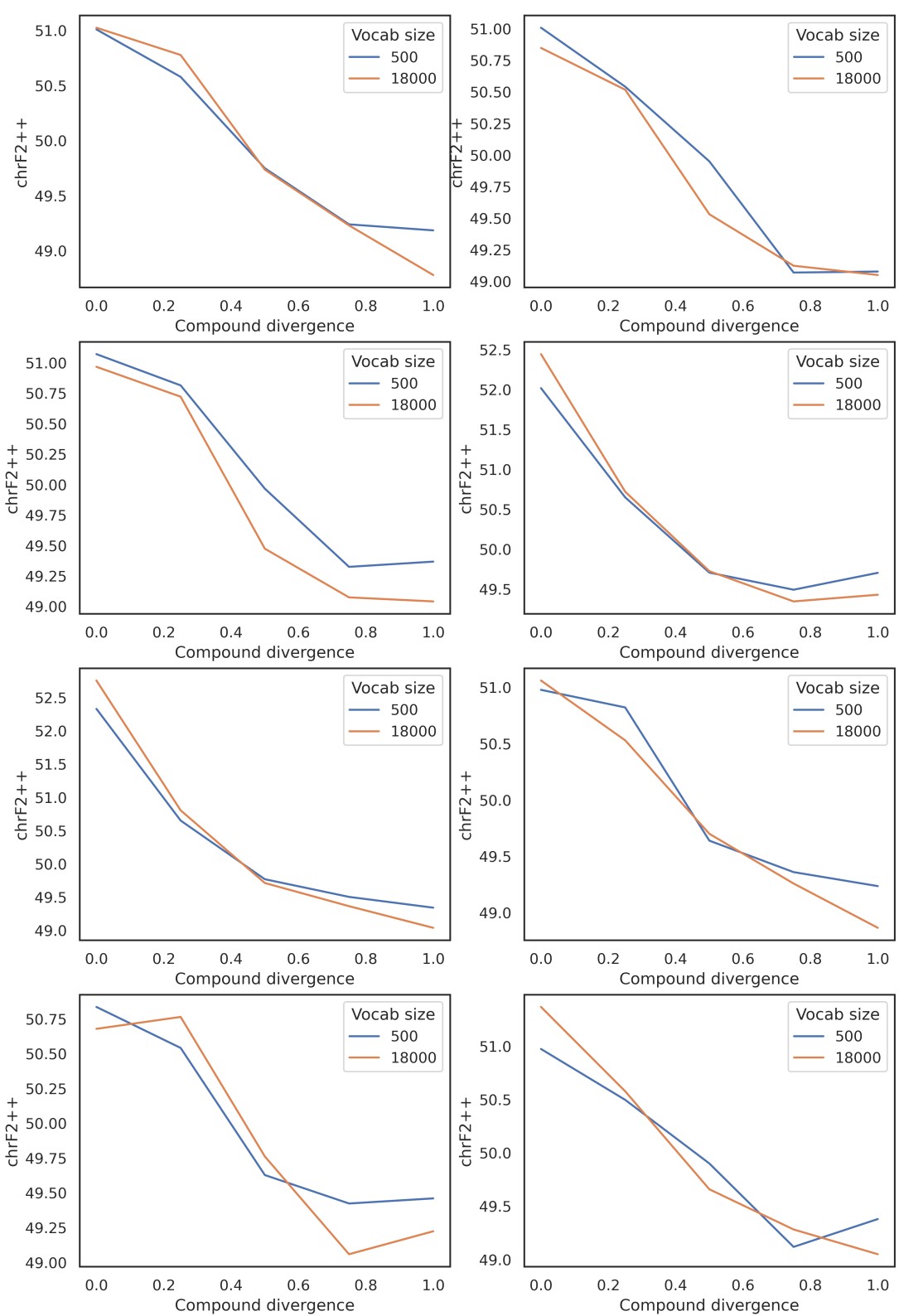

Figure 3: Comparison of vocabulary sizes 500 and 18000 with compound divergence values 0.0, 0.25, 0.5, 0.75 and 1.0. For 8 data split algorithm random seeds. The same results are partly in Table 4.