# OpenReview forum: "Evaluating Morphological Generalisation in Machine Translation by Distribution-Based Compositionality Assessment"
_NoDaLiDa/2023/Conference — NoDaLiDa 2023_

### Official Review · Reviewer_hY7a · 2023-03-10
**Evaluating morphological generalisation in machine translation by distribution-based compositionality assessment**

**Rating:** 8
**Confidence:** 2

**Review:**

This is a insightful and interesting paper with clear results and valuable contributions to sentence-based NLP.

**Paper Type:**

Long paper

---

### Official Review · Reviewer_QoA2 · 2023-03-10
**Nice paper on how to quantify the degree of morphological generalization in MT**

**Rating:** 8
**Confidence:** 3

**Review:**

This paper presents an approach to quantifying the degree of morphological generalization which is required in order to successfully generalize from an MT training set to a test set. The measure is by computing a divergence measure between the distribution of morphemes or morpheme compounds in the training and test sets.

The experimental results are somewhat modest but they do show correspondence between required generalization and MT performance. When the requirement for generalization is higher, models invariably do worse as expected. The experiments also show that models with smaller BPE vocabulary sizes tend to generalize better meaning that the performance drops less when the test set requires a higher degree of generalization.

A nice feature of this work, compared to earlier work, is that natural data is used to measure generalization performance in contrast to synthetic data.

**Paper Type:**

Long paper

---

### Official Review · Reviewer_kz4s · 2023-03-11
**Interesting application of a new, data-driven compositionality assessment method to NMT**

**Rating:** 8
**Confidence:** 4

**Review:**

This paper studies the ability of NMT to compositionally generalize between training and test samples, with a specific focus on morphological composition.
To this end, it adapts a recent compositionality assessment method (DBCA; Keysers et al. 2020) that is holistic, data-driven and based on naturally occurring data.
Briefly, the method consists of sampling training and test samples so that the distribution of 'compounds' (i.e. given combinations of morphological features) in the test is maximally divergent from the distribution of 'atoms' (i.e. lemmas) in the training data.

The assessment is applied to varying sizes of the source-language vocabulary in a Finnish-English MT task, where the size is determined by the number of BPE merges.

The results show that smaller subword vocabularies (i.e. less merges) correlate with better morphological generalisation.

The paper is very well written and structured. I really enjoy the read. It also represents an interesting, novel way to assess compositionality in MT that is complementary to previous efforts based on synthetic datasets.

I found the results unsurprising (it is pretty obvious to me that more segmentation implies a stronger ability to translate novel morphological combinations) and not particularly insightful (the vocabulary size that leads to best morphological compositionality is not the one that leads to the best overall MT quality: so concretely, what should one choose?)
It would have been interesting to at least make some hypotheses on which other aspects of translation may be hurt when shrinking the vocabulary size.

That said, the method and its application to MT remain valuable. Given the code will be released by the authors, this could be used by future work to study other variables in NMT, such as comparing BPE with alternative segmentation methods (while keeping the vocab size fixed)


RECOMMENDED EXTRA REFERENCE:

Choshen, Leshem, and Omri Abend. "Automatically Extracting Challenge Sets for Non-Local Phenomena in Neural Machine Translation." Proceedings of the 23rd Conference on Computational Natural Language Learning (CoNLL). 2019.

OTHER COMMENTS:
- Abstract & Intro: "generate an infinite number of novel meanings" => a potentially infinite
- Sect 3: the procedure to sample the list of lemmas (after filtering via the lists) seems rather complicated, but I don't fully understand why it is needed. Wouldn't random sampling lemma occurrences (not types) lead to a similar distribution?
- Tab 1: add English gloss



**Paper Type:**

Long paper

---

### Decision · Program_Chairs · 2023-03-17

Accept